# Protocol for assessment of the efficiency of CRISPR/Cas RNP delivery to different types of target cells

**Marina A. Tyumentseva** [ID]*[◎], **Aleksandr I. Tyumentsev**[◎], **Vasiliy G. Akimkin**[◎]

Central Research Institute of Epidemiology, Moscow, Russia

◎ These authors contributed equally to this work.
* tyumentseva@cmd.su

## Abstract

### Background

Delivery of CRISPR/Cas RNPs to target cells still remains the biggest bottleneck to genome editing. Many efforts are made to develop efficient CRISPR/Cas RNP delivery methods that will not affect viability of target cell dramatically. Popular current methods and protocols of CRISPR/Cas RNP delivery include lipofection and electroporation, transduction by osmocytosis and reversible permeabilization and erythrocyte-based methods.

### Methods

In this study we will assess the efficiency and optimize current CRISPR/Cas RNP delivery protocols to target cells. We will conduct our work using molecular cloning, protein expression and purification, cell culture, flow cytometry (immunocytochemistry) and cellular imaging techniques.

### Discussion

This will be the first extensive comparative study of popular current methods and protocols of CRISPR/Cas RNP delivery to human cell lines and primary cells. All protocols will be optimized and characterized using the following criteria i) protein delivery and genome editing efficacy; ii) viability of target cells after delivery (post-transduction recovery); iii) scalability of delivery process; iv) cost-effectiveness of the delivery process and v) intellectual property rights. Some methods will be considered 'research-use only', others will be recommended for scaling and application in the development of cell-based therapies.

## 1. Introduction

Efficacy and safety of genome editing and, consequently, of therapy is limited by the delivery of candidate molecules directly to target cells [1, 2]. Approaches to delivery of the therapeutic agent for genome editing can be divided into *in vivo* and *ex vivo* strategies.

**Funding:** This work is supported by the Ministry of Science and Higher Education of the Russian Federation within the framework of a grant in the form of a subsidy for the creation and development of the «World-class Genomic Research Center for

Ensuring Biological Safety and Technological Independence under the Federal Scientific and Technical Program for the Development of Genetic Technologies», agreement No. 075-15-2019-1666. The funders had and will not have a role in study design, data collection and analysis, decision to publish, or preparation of the manuscript.

**Competing interests:** The authors have declared that no competing interests exist.

The employed approach to delivery will depend on the type of a tool for genome editing. Genome-editing tools are delivered to a cell as genetically engineered constructs to accumulate the respective proteins within the cell. The delivery system must ensure highly efficient penetration of these gene constructs into the cell, resistance to degradation in the cell during transportation to the nucleus, and maintenance of the required level of expression. For example, when the *in vivo* strategy is employed, genome editing tools will be affected by the host immune system. The potential immune response will depend on the type of the delivery vehicle. Application of viral vectors can lead to long-term expression of hybrid nucleases, which, in its turn, can cause extensive damage to a human genome and a prolonged immune response.

Numerous viral and non-viral systems for delivery of genetically engineered constructs to cells of an organism have been developed [3, 4]. The most widely used viral systems are systems based on retroviruses, lentiviruses, adenoviruses, adeno-associated viruses, and the herpes simplex virus. For example, different serotypes of adeno-associated viruses enhance the efficiency of delivery to certain types of cells, thus making it tissue-specific [5]. The *in vivo* delivery of hybrid nucleases with adeno-associated viruses has been successfully used on different animal models involving metabolic disorders [6, 7], infection caused by the human immunodeficiency virus [8], muscular dystrophies [9], retinal diseases [10, 11], neurodegenerative diseases [12], *etc*.

As viral vectors contribute to the efficient delivery and the longer-lasting expression of genome editing tools, they are increasingly promising for clinical application. However, immune responses induced by viral delivery systems can be a crucial factor in limiting the therapeutic potential of the delivered construct for genome editing [13, 14]. It turns into a major challenge when genome editing implies repeated (or multiple and long-term) administration of a therapeutic gene product. Some of these limitations can be overcome with combined immunosuppressive treatment [15].

Non-viral systems involve direct administration of DNA/RNA to cells and tissues by using electroporation, liposomes, cationic polymers, *etc*. [16]. Recently developed lipid-based nanoparticles have been approved for therapeutic application [17]; gold nanoparticles were successfully used in rodent models to treat Duchenne muscular dystrophy [18] and fragile X syndrome [19].

Similar delivery systems are also used *ex vivo* for precise genome editing. Depending on the type of cells used in *ex vivo* delivery, tools for genome editing can be delivered both through viral vectors and by using electroporation, microinjections, cell-penetrating peptides or nanoparticles. Pluripotent stem cells are also popular in *ex vivo* genome editing applications [20]. Induced pluripotent stem cells (iPSCs) have attracted considerable interest as promising model systems, as they can be differentiated into any type of cells relating to the studied disease, for example, into skeletal muscle cells [21–23], hepatocytes [24, 25], cardiac muscle cells [26, 27], and many others.

Each of these delivery formats has its advantages and drawbacks; therefore, in most cases, preference is given to a combination of viral and non-viral systems [28]. Besides, it has been demonstrated that some genome-editing tools possess the innate ability to cross cell membranes and induce a targeted gene knockout in human cells [29]. In direct delivery system, the off-target DNA cleavage rates were significantly lower than the rates observed for expression in the cell. Lower rates of off-target genome cleavage can be caused by the shorter time, during which the nuclease stays in the cell. When this approach is applied, high rates of the gene knockout can be reached only after repeated treatment of cells, thus substantially limiting the application of this technique for *ex vivo* genome editing. Nevertheless, incorporation of tandem NLS repeats (where NLS stands for 'nuclear localization signal') into the nuclease backbone can increase its cell-penetrating activity by up to 13 times [30]. Furthermore, even one-

time treatment enhanced the efficiency of the gene knockout in many types of human cells, including CD4+ T cells and iPSCs.

The efficient delivery of CRISPR/Cas genome editing elements to target cells is of paramount importance for using CRISPR/Cas tools in therapy [31]. Generally, three strategies of delivery of CRISPR/Cas elements are used—*in vitro*, *ex vivo* and *in vivo*. CRISPR/Cas are commonly delivered using physical methods, viral and non-viral vector delivery, *etc*. Physical methods of delivery imply short-term disruption of the target cell membrane and include electroporation, sonoporation, nano-injection, microinjection, and hydrodynamic injection [32]. Viral vectors are the earliest molecular tool for transfer of genes to human cells; they transfer nucleic acids encoding CRISPR/Cas components to target cells in the envelope of a virus, for example, an adenovirus, adeno-associated virus, retrovirus, lentivirus, Epstein–Barr virus, herpes simplex virus and bacteriophages [33, 34]. In addition, alternative (non-viral) methods of CRISPR/Cas delivery, for example, by using lipid nanoparticles, polymer nanoparticles and hydrogel nanoparticles, hybrid gold, graphene oxide, metal-organic frameworks, black phosphorus nanomaterials, *etc*. were reported [35].

CRISPR/Cas elements can be delivered to a living cell as a set of plasmid DNAs encoding the Cas protein and guide RNA or as a combination of the Cas-protein-encoding mRNA and the guide RNA. The third option suggests delivery of the pre-assembled ribonucleoprotein complex (Cas protein and the guide RNA) into the cell. The CRISPR/Cas delivery in the form of a ribonucleoprotein complex has several advantages, including high editing efficiency; low non-specific activity; editing starts immediately after the delivery to the cell; fast screening of efficiency of guide RNAs in vitro; reduced immunogenicity due to the short-term presence of CRISPR/Cas elements in the target cell. Therefore, ribonucleoprotein complexes offer promising opportunities in CRISPR/Cas-based genome editing.

To date, many strategies are available for CRISPR/Cas RNP delivery based on physical approaches and synthetic carriers. CRISPR/Cas RNP were successfully delivered to cells using microinjection [36], biolistics [37, 38], electroporation [39–43], microfluidics [44, 45], filtroporation [46], nanotube [47], osmocytosis [48], synthetic lipid nanoparticles [49], cell penetrating peptides (CPPs) [50], lipopeptides [51], dendrimers [52], chitosan nanoparticles [53], nanogels [54], gold nanoparticles [55], metal-organic frameworks [56], graphene oxide [57], black phosphorus nanosheets [58], calcium phosphate nanoparticles [59], and many more [51].

Nevertheless, delivery of CRISPR/Cas RNPs to target cells still remains some kind of problem to genome editing in clinical practice and drug (therapy) development. Several methods are developed for efficient CRISPR/Cas RNP delivery. But there is no data on comparison of the most popular current methods and protocols used for CRISPR/Cas RNP delivery to human cell lines and primary cells. This article represents a study protocol of the first extensive comparative study of current methods of CRISPR/Cas RNP delivery to target cells and focuses on the following criteria i) protein delivery and genome editing efficacy; ii) viability of target cells after delivery (post-transduction recovery); iii) scalability of delivery process; iv) cost-effectiveness of the delivery process and v) intellectual property rights. Some methods will be considered 'research-use only', others will be recommended for scaling and application in the development of cell-based therapies (Fig 1).

## 2. Materials and methods

- Study aim

The study aim is to assess the efficiency of CRISPR/Cas RNP delivery protocols to target cells and to optimize these protocols.

| Cell-based therapy: | Research-use only: |
|---|---|
| ✓ Moderate to high protein delivery efficacy; | ✓ Low to moderate protein delivery efficacy; |
| ✓ Moderate to high genome editing efficacy; | ✓ Low to moderate genome editing efficacy; |
| ✓ Moderate to high post-transduction recovery rate; | ✓ Low to moderate post-transduction recovery rate; |
| ✓ Scalable; | ✓ Not scalable; |
| ✓ Cost-effective; | ✓ Expensive; |
| ✓ IP-free. | ✓ Third party IP. |

**Fig 1. Features of CRISPR/Cas RNP delivery methods suitable for development of cell-based therapies and those considered 'research-use only'.**

- Study objectives

    i. To estimate the efficiency of CRISPR/Cas RNP delivery to immortalized adherent and suspension cell lines.

    ii. To estimate the efficiency of CRISPR/Cas RNP delivery to human primary T-cells and CD34+-cells.

    iii. To optimize CRISPR/Cas RNP delivery protocol to immortalized adherent and suspension cell lines and primary human cells.

- Study design and setting

    We will conduct a comparative study of various methods of CRISPR/Cas RNP delivery to mammalian cells and will optimize these protocols.

- Study methods

    The study will be conducted using the following methods:

a. molecular cloning;

b. protein expression and purification;

c. cell culture;

d. flow cytometry (immunocytochemistry);

e. cellular imaging including real-time cell analysis.

- Study outcomes

    After the study completion we will possess the data regarding i) protein delivery and genome editing efficacy; ii) post-transduction recovery; iii) scalability; iv) cost-effectiveness and v) intellectual property burden on the CRISPR/Cas RNP delivery methods to target cells. The methods used will include commercially available transfection reagents, electroporation,

transduction by osmocytosis and reversible permeabilization. Also, we will provide optimized protocols for CRISPR/Cas RNP delivery to different cell types (adherent and suspension cells, primary human cells) with enhanced genome editing efficacy.

• Study details

## 2.1. Molecular cloning, expression and purification of eGFP-tagged CRISPR/Cas proteins, CRISPR/Cas RNP assembly

CRISPR/Cas proteins with C-terminal eGFP tag will be constructed from plasmids encoding CRISPR/Cas proteins (SpCas9, SpD10ACas9, STCas9, AsCpf1 and LbCpf1) obtained earlier [60].

Expression of CRISPR/Cas proteins will be conducted in the E. coli Rosetta-gami B (DE3). Purification of CRISPR/Cas proteins will be carried out using Chelating Sepharose FF (GE Healthcare) and SP Sepharose FF (GE Healthcare) [60].

Quality control will be performed for all purified CRISPR/Cas proteins. Aggregate analysis will be performed using high resolution gel filtration chromatography on Superdex® 200 Increase 10/300 GL (Cytiva) using ÄKTA avant 25 (Cytiva). Bacterial endotoxins in CRISPR/ Cas protein preparations will be measured utilizing Pierce™ Chromogenic Endotoxin Quant Kit (Thermo Fisher Scientific). If necessary, CRISPR/Cas proteins will be additionally purified as described earlier [60]. All CRISPR/Cas protein preparations will be filter-sterilized 0.22 and microbiologically tested.

Validated gRNA sequences that have demonstrated high level editing efficiency (up to 90%) will be used in all experiments (AAVS1, CDK4 and HPRT1; Thermo Fisher Scientific); non-targeting gRNA sequence that do not recognize any sequence in the human genome will serve as non-edited control when performing analysis on pooled populations of edited cells.

Guide RNAs synthesis will be performed in the in vitro transcription (IVT) reaction (HiScribe™ T7 High Yield RNA Synthesis Kit, NEB), IVT products will be purified from the reaction mixture by adding of sodium chloride (400 mM) and an equal volume of isopropyl alcohol. Ribonucleoprotein complex assembly will be carried out according to a standard protocol with slight modifications [61].

Nuclease activity of the assembled CRISPR/Cas RNPs will be evaluated using supercoiled plasmid with target insert [60]. Efficient hydrolysis will lead to plasmid linearization or relaxation which will result in mobility changes on electrophoresis (hydrolysis products are visualized by electrophoresis in agarose gel). Semi-quantitative assessment of the hydrolysis rate will be carried out using software of gel-documentation system (Bio-Rad GelDoc XR+ and/or Vilber Fusion-FX7).

## 2.2. CRISPR/Cas RNP delivery to target cells

CRISPR/Cas RNPs will be delivered to target cells using either commercially available transfection reagents, electroporation, transduction by osmocytosis, reversible permeabilization or erythrocyte-based methods. For each delivery technique the optimal ratio (1:1, 1:2 or 1:3) of gRNA to CRISPR/Cas protein required for precomplexing will be determined. Afterwards, the optimal amount of recombinant CRISPR/Cas will be determined. When optimal conditions will be selected, each RNP delivery protocol will be optimized to result in maximal target gene knockout (KO).

**2.2.1. Cells.** Adherent and suspension cell lines will be purchased from The European Collection of Authenticated Cell Cultures (ECACC) repository (Merck) and the Centre for AIDS Reagents (CFAR) repository (NIBSC).

Primary human T-cells (CD4+/CD8+) and CD34+-cells will be obtained on CliniMACS Prodigy® Instrument (Miltenyi Biotec) using CliniMACS® CD4, CD8 and CD34 Product Line reagents (Miltenyi Biotec). Leukopak (e.g., leukocyte concentrate) will be purchased from State Budgetary Healthcare Institution of the Moscow Region "Moscow Regional Blood Transfusion Station".

**2.2.2. Delivery using transfection reagents.** CRISPR/Cas RNPs will be delivered to target cells using either Lipofectamine™ CRISPRMAX™ Cas9 Transfection Reagent (Thermo Fisher Scientific), TransIT-X2® Dynamic Delivery System (Mirus Bio LLC) or branched PEI (MilliporeSigma).

*2.2.2.1. Lipofectamine™ CRISPRMAX™ complex formation.* 5 μl of Opti-MEM medium will be added to a sterile 0.6 ml tube, followed by the addition of CRISPR/Cas nuclease (to final concentration 6 nM, 12 nM, or 24 nM) and IVT gRNA in a 1:1, 1:2 or 1:3 molar ratio. Upon mixing by vortexing briefly, 0.2 μl Cas9 Plus™ Reagent (the ratio of CRISPR/Cas nuclease to Cas9 Plus™ Reagent is 1:2 (μg:μL)) will be added to the solution containing CRISPR/Cas protein and gRNA. After vortexing, the mixture will be incubated at 25˚C for 5 min to allow the formation of CRISPR/Cas RNPs. In the meantime, 5 μl Opti-MEM medium will be added to a separate sterile tube, followed by addition of 0.3 μl of Lipofectamine™ CRISPRMAX™. After briefly vortexing, the Lipofectamine™ CRISPRMAX™ solution will be incubated at 25˚C for approx. 3–5 min. After incubation, the CRISPR/Cas RNPs will be then added to the Lipofectamine™ CRISPRMAX™ solution. Upon mixing, the sample will be incubated at 25˚C for 5–10 min to form CRISPR/Cas RNPs and Lipofectamine™ CRISPRMAX™ complexes.

*2.2.2.2. TransIT-X2® complex formation.* 10 μl of Opti-MEM medium will be added to a sterile 0.6 ml tube, followed by the addition CRISPR/Cas nuclease (to final concentration 6 nM, 12 nM, or 24 nM) and IVT gRNA in a 1:1, 1:2 or 1:3 molar ratio. Upon mixing gently by pipetting, 0.5 μl TransIT-X2® will be added to the solution containing CRISPR/Cas protein and gRNA. After mixing, the solution will be incubated at 25˚C for 15 min to allow the formation of CRISPR/Cas RNPs and TransIT-X2® complexes.

*2.2.2.3. Branched PEI complex formation.* After formation of CRISPR/Cas RNPs (CRISPR/Cas nuclease to final concentration 6 nM, 12 nM, or 24 nM and IVT gRNA in a 1:1, 1:2 or 1:3 molar ratio), PEI (Polyethylenimine, branched average Mw ~25,000 by LS, average Mn ~10,000 by GPC, branched) will be added to final concentration 60 nM, 120 nM, or 240 nM to the RNPs and vortexed. The resulting solution (i.e., CRISPR/Cas RNPs and PEI complex) will be incubated for 20 minutes at 25˚C. After incubation Opti-MEM medium to final volume of 100 μl will be added and tubes will be incubated at 37˚C for 20 minutes to temperature equilibrate with cells to be transfected.

*2.2.2.4. Forward transfection protocol.* One day prior to transfection, cells will be plated in complete growth medium to be 30–80% confluent at the time of transfection (most adherent cell types are seeded at $0.8–1.8×10^5$/ml). Schematic layout of transfection variations used for assessment of CRISPR/Cas RNPs delivery in 96-well plate is given in Fig 2.

On the day of transfection conditioned culture media will be aspirated and transfection complexes will be added drop-wise to different areas of the well. Final volume will be adjusted to 100 μl with complete culture media.

*2.2.2.5. Reverse transfection protocol.* For all target cells (adherent, suspension and primary), a "reverse transfection" protocol where freshly passaged cells are used for transfection will be applied. Transfection plate layout will be the same as for forward transfection (Fig 2). On the day of transfection CRISPR/Cas RNP transfection complexes will be pre-plated to the wells of 96-well plate according to transfection plate layout afterwards, $2.0–5.0×10^4$ cells per well will be added.

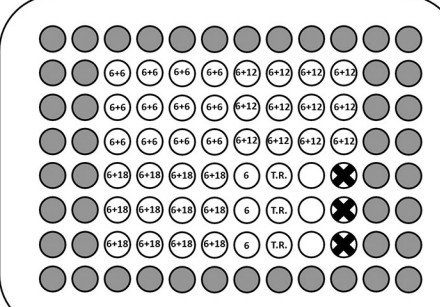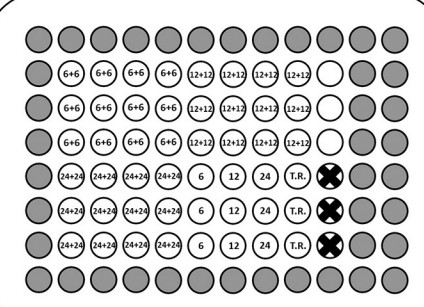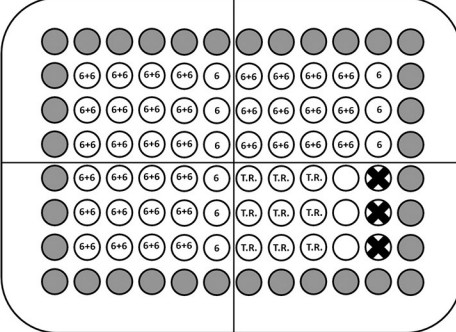

**Fig 2. Example of transfection plate layout.** Experiments where (A) optimal ratio of gRNA to CRISPR/Cas protein required for precomplexing will be determined, (B) the optimal amount of recombinant CRISPR/Cas will be determined (if we assume the optimal ratio will be 1:1) and (C) optimization of transfection reagent amount (1×, 2× and 3×) used for complex formation (if we assume the optimal CRISPR/Cas amount is 0.6 pmol per well). Wells named "T.R." will contain transfection reagent only. White wells will hold non-transfected cells used as non-treated controls for the cell viability assay. The crossed-out wells represent empty wells used for holding the PrestoBlue™ Cell Viability Reagent staining blank. Grey wells will be filled with PBS to avoid edge effect. All experiments will be performed in triplicate, 4 gRNAs will be used for RNP formation (AAVS1, CDK4 and HPRT1 and non-targeting control gRNA).

*2.2.2.6. Modified reverse transfection protocol.* For "modified reverse" transfections, adherent cells will be passaged and plated immediately before transfection complexes are added to the cells. In this case, cells will be loosely adhered to the plate surface by the time they interact with the transfection complexes.

24 hours prior to transfection, $2.0–4.0\times10^6$ cells will be plated in a T-75 cm2 flask so that they will be 70–80% confluent the following day (approximately $2.0–6.0\times10^6$ cells will be needed per 96-well plate). On the day of transfection (< 1 hour prior to transfection) cells will be trypsinized and counted using Trypan Blue stain (0.4%) on Countess® II FL Automated Cell Counter (Thermo Fisher Scientific) to determine the appropriate volume of cells in media to obtain $1.6–4.8\times10^5$ cells per ml. 50 μl of diluted cell mixture ($0.8–2.4\times10^4$ cells) will be added to each well and gently rocked back and forth and from side to side to distribute the cells evenly. Immediately after, transfection complexes will be added drop-wise to different areas of the well according to transfection plate layout (Fig 2) and final volume will be adjusted to 100 μl with complete growth media.

All experiments will be performed in triplicate, data will be presented as mean ± SD (n = 3) and statistical analysis will be performed by one-way ANOVA.

**2.2.3. CRISPR/Cas RNP delivery to target cells using electroporation.** CRISPR/Cas RNPs will be delivered to target cells using either 4D-Nucleofector™ System (Lonza) or Neon™ Transfection System (Thermo Fisher Scientific).

*2.2.3.1. CRISPR/Cas RNP delivery to target cells using 4D-Nucleofector™ System.* Prior to CRISPR/Cas RNPs Nucleofection™ cells will be subcultured 1–2 days. On the day of Nucleofection™ cell confluency must be 70–85% as higher cell densities may cause lower Nucleofection™ Efficiencies. $1–5\times10^5$ adherent cells or $0.2–1\times10^6$ suspension cells will be used for 20 μl Nucleocuvette™ Strip.

First, we will determine the optimal ratio (1:1, 1:2 or 1:3) of gRNA to Cas protein required for precomplexing that will result in maximal target gene KO. The amount of Cas protein will be constant at 5 μg (30 pmol or 1.5 μM) for these experiments. We will use the Lonza 4D-Nucleofector™ System with conditions recommended by the manufacturer for plasmid transfection of target cells (for example, pulse EO-115 and buffer P3 for primary human T cells or pulse EO-100 and buffer P3 for primary human CD34+ cells). Next, we will determine the optimal amount of recombinant CRISPR/Cas (1.5 μM, 3 μM or 6 μM) protein. Finally, the optimization experiment for immortalized cell lines will comprise three different Cell Line

4D-Nucleofector™ Solutions SE, SF, SG (Lonza) and Ingenio® Electroporation Solution (Mirus Bio LLC). For primary human cells five different 4D-Nucleofector™ Solutions P1, P2, P3, P4 and P5 (Lonza), Ingenio® Electroporation Solution (Mirus Bio LLC) and home-made Sol2 [40] and solution V [62] will be used for optimization of delivery protocol. Each solution will be tested in combination with 15 different Nucleofector™ Programs plus one control (see Table 1). Example of experiment layout is given on Fig 3.

After Nucleofection™ run completion, Nucleocuvette™ Strip will be carefully removed from the retainer and incubated for 10 min at room temperature. Then cells will be resuspended with pre-warmed medium (80 μL), mixed by gently pipetting and plated for further analysis (25 μL of cell suspension after adherent cells Nucleofection™ procedure and 50 μL - after suspension cells Nucleofection™ procedure per well of 96-well plate). All experiments will be performed in duplicate, data will be presented as mean ± SD (n = 2), statistical analysis will be performed by one-way ANOVA.

*2.2.3.2. CRISPR/Cas RNP delivery to target cells using Neon™ Transfection System*. Prior to Neon™ electroporation cells will be subcultured for 2 days. On the day before transfection cells will be seeded so that they will be 30–70% confluent for transfection. $0.5–2 \times 10^6$ adherent cells or $1–5 \times 10^6$ suspension cells will be resuspended in the 100 μL of appropriate buffer containing CRISPR/Cas RNPs and transferred to Neon™ Tip.

The optimal ratio of gRNA to Cas protein and the amount of Cas protein will be determined as described earlier (see 2.3.1) using Neon™ Transfection System with conditions recommended by the manufacturer for plasmid transfection of target cells (for example, 1600 V/10 ms/3 pulses and Buffer R for primary human T cells). For further optimization resuspension Buffers R and T (Thermo Fisher Scientific), Ingenio® Electroporation Solution (Mirus Bio LLC) and home-made Sol2 [40] and solution V [62] will be used. Each solution will be tested in combination with 23 different preprogrammed Neon™ protocols plus 1 control (see Table 2).

After Neon™ electroporation, cells will be transferred to pre-warmed culture medium (2 mL) and plated for further analysis (6-well plate). All experiments will be performed in duplicate, 4 gRNAs will be used for RNP formation (AAVS1, CDK4 and HPRT1 and non-targeting control gRNA). Data will be presented as mean ± SD (n = 2), statistical analysis will be performed by one-way ANOVA.

**2.2.4. CRISPR/Cas RNP delivery using transduction by osmocytosis.** CRISPR/Cas RNPs will be delivered to adherent cell lines using iTOP technique [63], GSM method [64] or standard osmotic lysis of pinocytic vesicles technique [65].

**Table 1. CRISPR/Cas RNP delivery to immortalized cell lines and primary human cells using 4D-Nucleofector™ System.**

| Immortalized cell lines | | Primary human cells | |
|---|---|---|---|
| Solution SE/SF/SG/Ingenio® Electroporation Solution | | Solution P1/P2/P3/P4/P5/Ingenio® Electroporation Solution/Solution V/Sol2 | |
| CA-137 | DS-150 | CA-137 | DS-150 |
| CM-138 | DS-120 | CM-138 | DS-120 |
| CM-137 | EH-100 | CM-137 | EH-100 |
| CM-150 | EO-100 | CM-150 | EO-100 |
| DN-100 | EN-138 | DN-100 | EN-138 |
| DS-138 | EN-150 | DS-138 | EN-150 |
| DS-137 | EW-113 | DS-137 | EW-113 |
| DS-130 | Control | DS-130 | Control |

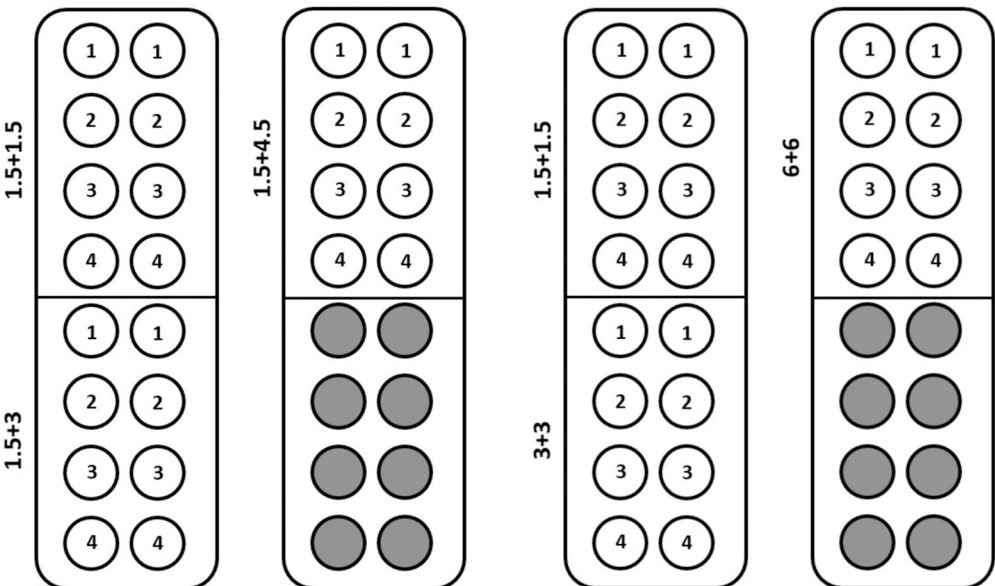

**Fig 3. Example of 20 µl Nucleocuvette™ Strip layout.** Experiments where (A) optimal ratio of gRNA to CRISPR/Cas protein required for precomplexing will be determined, (B) the optimal amount of recombinant CRISPR/Cas will be determined (if we assume the optimal ratio will be 1:1). All experiments will be performed in duplicate, 4 gRNAs will be used for RNP formation (AAVS1, CDK4 and HPRT1 and non-targeting control gRNA).

For each method mentioned above cells will be subcultured before transduction for 1–2 days. On the day before transduction from $2.5 \times 10^4$ to $5.0 \times 10^4$ cells/well will be seeded in 48-well culture plate.

First of all, optimal gRNA to Cas protein ratio (1:1, 1:2 or 1:3) will be determined using 250 pmol (1 µM) of Cas protein in transduction mixture. The composition of transduction mixture for iTOP technique (complete transduction mixture is Opti-MEM culture media supplemented with 5 mM NaH2PO4, 425 mM NaCl, 30 mM glycerol, 15 mM glycine, 0.25 mM MgCl2, 0.2 mM β-mercaptoethanol, 200 mM GABA, 1×NEAA, 2 mM L-glutamine), GSM method (GSM transduction mixture is Opti-MEM culture media supplemented with 600 mM glucose, 600 mM sucrose and 600 mM mannitol) and standard osmotic lysis of pinocytic vesicles technique (500 mM sucrose, 10% PEG1000 then rinsed with hypotonic solution containing six parts of culture media and four parts of mQ water) will correspond to that published in [63–65] respectively. The optimal amount of recombinant CRISPR/Cas protein (1 µM, 2 µM or 4 µM) for each transduction protocol will also be assessed. Osmolality of all transduction mixtures will be controlled using KNAUER Semi-Micro Osmometer K-7400S (KNAUER Wissenschaftliche Geräte GmbH).

*2.2.4.1. CRISPR/Cas RNP delivery to target cells using modified iTOP technique.* Further optimization of iTOP technique will include (i) optimization of key iTOP components and (ii) optimization of osmoprotectants. As it was shown iTOP protein transduction was directly proportional to the NaCl-induced hyperosmolality and GABA yielded excellent protein transduction efficiency and minimal effect on cell proliferation [63] we will optimize concentrations of these key iTOP components. Briefly, 250 µL of Opti-MEM culture media with 5 mM NaH2PO4, from 0 to 595 mM NaCl, 30 mM glycerol, 15 mM glycine, 0.25 mM MgCl2, 0.2 mM β-mercaptoethanol, from 0 to 250 mM GABA, 1×NEAA and 2 mM L-glutamine containing CRISPR/Cas RNP will be added to corresponding well of cell culture plate (layout is represented in Table 3).

**Table 2. CRISPR/Cas RNP delivery to target cells using Neon™ Transfection System.**

| | Resuspension Buffer R/Resuspension Buffer T/Ingenio® Electroporation Solution/ Solution V/Sol2 | | |
|---|---|---|---|
| | Pulse voltage, V | Pulse width, ms | Pulse number |
| **1** | 1400 | 20 | 1 |
| **2** | 1500 | 20 | 1 |
| **3** | 1600 | 20 | 1 |
| **4** | 1700 | 20 | 1 |
| **5** | 1100 | 30 | 1 |
| **6** | 1200 | 30 | 1 |
| **7** | 1300 | 30 | 1 |
| **8** | 1400 | 30 | 1 |
| **9** | 1000 | 40 | 1 |
| **10** | 1100 | 40 | 1 |
| **11** | 1200 | 40 | 1 |
| **12** | 1100 | 20 | 2 |
| **13** | 1200 | 20 | 2 |
| **14** | 1300 | 20 | 2 |
| **15** | 1400 | 20 | 2 |
| **16** | 850 | 30 | 2 |
| **17** | 950 | 30 | 2 |
| **18** | 1050 | 30 | 2 |
| **19** | 1150 | 30 | 2 |
| **20** | 1300 | 10 | 3 |
| **21** | 1400 | 10 | 3 |
| **22** | 1500 | 10 | 3 |
| **23** | 1600 | 10 | 3 |

After 1, 2, 3 and 12 hr, the complete transduction mixture will be replaced by fresh cell culture media and cells will be incubated for 84 hr for further analysis.

Osmoprotectants in the iTOP transduction buffer effectively prevent hypertonicity-induced DNA damage [63]. So, it will be very important to optimize concentrations of these osmoprotectants. To do this 250 µL of Opti-MEM culture media with 5 mM $NaH_2PO_4$, optimal concentration of NaCl, from 0 to 1360 mM glycerol, from 0 to 135 mM glycine, 0.25 mM $MgCl_2$, 0.2 mM β-mercaptoethanol, from 0 mM to 250 mM GABA, 1×NEAA, 2 mM L-glutamine and CRISPR/Cas RNP will be added to corresponding well of cell culture plate (layout is represented in Table 4).

In addition to glycine, betaine and proline are often used as osmoportectors [66]. Therefore, we will optimize concentrations of these osmoprotectants too as described earlier with transduction plate layout represented in Tables 5 and 6.

To estimate CRISPR/Cas RNP toxicity a mock transduction (i.e., iTOP transduction mixture only) will be included to each experiment.

After 1, 2, 3 and 12 hr, the complete transduction mixture will be replaced by fresh cell culture media and cells will be incubated for 84 hr. All experiments will be performed in triplicate, 4 gRNAs will be used for RNP formation (AAVS1, CDK4 and HPRT1 and non-targeting control gRNA). Data will be presented as mean ± SD (n = 3), statistical analysis will be performed by one-way ANOVA.

*2.2.4.2. CRISPR/Cas RNP delivery to target cells using modified GSM method.* In addition to the iTOP method, the GSM method (where "G" stands for glucose, "S"–for sucrose, and "M"–

**Table 3. NaCl and GABA concentrations optimization during iTOP protein transduction (48-well transduction plate layout).**

| | | | | NaCl (Stock solution 5 M) | | | | | | | |
| --- | --- | --- | --- | --- | --- | --- | --- | --- | --- | --- | --- |
| | | | | Concentration in Complete transduction mixture, mM | | | | | | | |
| | | | | 0 | 85 | 170 | 255 | 340 | 425 | 510 | 595 |
| | | | | 1 | 2 | 3 | 4 | 5 | 6 | 7 | 8 |
| GABA (Stock solution 3,3 M) | Concentration in **Complete transduction mixture, mM** | 0 | A | | | | | | | | |
| | | 50 | B | | | | | | | | |
| | | 100 | C | | | | | | | | |
| | | 150 | D | | | | | | | | |
| | | 200 | E | | | | | | * | | |
| | | 250 | F | | | | | | | | |

* - optimal combination according to [63].

for mannitol) is used to deliver proteins into the cells. GSM protein transduction is known to be directly proportional to the concentrations of glucose, sucrose and mannitol [64]. So, in our future study we will optimize concentrations of these key GSM components. For glucose, sucrose and mannitol concentrations optimization refer to Table 7. Corresponding concentration will be added to Opti-MEM culture media with CRISPR/Cas RNP with final volume 250 μL.

Similarly to iTOP method, after 1, 2, 3 and 12 hr, the complete transduction mixture will be replaced by fresh cell culture media and cells will be incubated for 84 hr. All experiments will be performed in triplicate, 4 gRNAs will be used for RNP formation (AAVS1, CDK4 and HPRT1 and non-targeting control gRNA). Data will be presented as mean ± SD (n = 3), statistical analysis will be performed by one-way ANOVA.

*2.2.4.3. CRISPR/Cas RNP delivery to target cells using standard osmotic lysis of pinocytic vesicles technique.* During optimization of standard osmotic lysis of pinocytic vesicles technique [65] we will treat cells with transduction solution: 250 μL Opti-MEM containing CRISPR/Cas RNP and from 200 to 600 mM sucrose and 10% PEG1000 (optimization experiment setup is given in Table 8). Transduction will last 5, 10, 15 and 20 min and then cells will be rinsed with hypotonic solution containing six parts of culture media and four parts of mQ water.

A mock transduction control will be included to understand CRISPR/Cas RNP toxicity when compared to the non-treated sample. All experiments will be performed in triplicate, 4

**Table 4. Osmoprotectants concentrations optimization during iTOP protein transduction (48-well transduction plate layout).**

| | | | | Glycerol (Stock solution 6.85 M or 50%) | | | | | | | |
| --- | --- | --- | --- | --- | --- | --- | --- | --- | --- | --- | --- |
| | | | | Concentration in Complete transduction mixture, mM (%) | | | | | | | |
| | | | | 0 (0%) | 10 (0.07%) | 30 (0.22%) | 90 (0.66%) | 270 (1.98%) | 810 (5.94%) | 1360 (10%) | |
| | | | | 1 | 2 | 3 | 4 | 5 | 6 | 7 | 8 |
| Glycine (Stock solution 2.5 M) | Concentration in **Complete transduction mixture, mM** | 0 | A | | | | | | | | - |
| | | 5 | B | | | | | | | | - |
| | | 15 | C | | | * | | | | | - |
| | | 45 | D | | | | | | | | - |
| | | 135 | E | | | | | | | | - |
| | | | F | - | - | - | - | - | - | - | - |

* - optimal combination according to [63]. Row "F" and column #8 will be filled with PBS.

**Table 5. Betaine and glycerol concentrations optimization during iTOP protein transduction (48-well transduction plate layout).**

| | | | | Glycerol (Stock solution 6.85 M or 50%) | | | | | | | |
| | | | | Concentration in Complete transduction mixture, mM (%) | | | | | | | |
| | | | | 0 (0%) | 10 (0.07%) | 30 (0.22%) | 90 (0.66%) | 270 (1.98%) | 810 (5.94%) | 1360 (10%) | |
| | | | | 1 | 2 | 3 | 4 | 5 | 6 | 7 | 8 |
| Betaine (Stock solution 5 M) | Concentration in **Complete transduction mixture, mM** | 0 | A | | | | | | | | - |
| | | 10 | B | | | | | | | | - |
| | | 30 | C | | | | | | | | - |
| | | 90 | D | | | | | | | | - |
| | | 270 | E | | | | | | | | - |
| | | | F | - | - | - | - | - | - | - | - |

Row "F" and column #8 will be filled with PBS.

gRNAs will be used for RNP formation (AAVS1, CDK4 and HPRT1 and non-targeting control gRNA). Data will be presented as mean ± SD (n = 3), statistical analysis will be performed by one-way ANOVA.

**2.2.5. CRISPR/Cas RNP delivery to target cells using reversible permeabilization methods.** O'Dea et al. have shown that proteins can be efficiently delivered to cells using the delivery solution (DS) which contains 32 mM sucrose, 12 mM potassium chloride, 12 mM ammonium acetate, 5 mM HEPES and 25% ethanol [67]. The optimal gRNA to CRISPR/Cas protein ratio (1:1, 1:2 or 1:3) and optimal CRISPR/Cas protein amount (1 μM, 2 μM or 4 μM) will be determined using published recipe of DS. We will make an attempt to optimize this protocol for CRISPR/Cas RNP delivery varying concentrations of components of the DS:

- 0–150 mM sucrose;

- 0–50 mM potassium chloride;

- 0–50 mM ammonium acetate;

- 0–25 mM HEPES;

- 0–93% ethanol.

**Table 6. Proline and glycerol concentrations optimization during iTOP protein transduction (48-well transduction plate layout).**

| | | | | Glycerol (Stock solution 6.85 M or 50%) | | | | | | | |
| | | | | Concentration in Complete transduction mixture, mM (%) | | | | | | | |
| | | | | 0 (0%) | 10 (0.07%) | 30 (0.22%) | 90 (0.66%) | 270 (1.98%) | 810 (5.94%) | 1360 (10%) | |
| | | | | 1 | 2 | 3 | 4 | 5 | 6 | 7 | 8 |
| Proline (Stock solution 2 M) | Concentration in **Complete transduction mixture, mM** | 0 | A | | | | | | | | - |
| | | 6.66 | B | | | | | | | | - |
| | | 20 | C | | | | | | | | - |
| | | 60 | D | | | | | | | | - |
| | | 180 | E | | | | | | | | - |
| | | | F | - | - | - | - | - | - | - | - |

Row "F" and column #8 will be filled with PBS.

**Table 7. Glucose, sucrose and mannitol concentrations optimization for GSM protein transduction.**

| Well | Glucose, mM | Sucrose, mM | Mannitol, mM |
|---|---|---|---|
| 1 | 200 | 200 | 200 |
| 2 | 200 | 200 | 400 |
| 3 | 200 | 200 | 600 |
| 4 | 200 | 400 | 200 |
| 5 | 200 | 400 | 400 |
| 6 | 200 | 400 | 600 |
| 7 | 200 | 600 | 200 |
| 8 | 200 | 600 | 400 |
| 9 | 200 | 600 | 600 |
| 10 | 400 | 200 | 200 |
| 11 | 400 | 200 | 400 |
| 12 | 400 | 200 | 600 |
| 13 | 400 | 400 | 200 |
| 14 | 400 | 400 | 400 |
| 15 | 400 | 400 | 600 |
| 16 | 400 | 600 | 200 |
| 17 | 400 | 600 | 400 |
| 18 | 400 | 600 | 600 |
| 19 | 600 | 200 | 200 |
| 20 | 600 | 200 | 400 |
| 21 | 600 | 200 | 600 |
| 22 | 600 | 400 | 200 |
| 23 | 600 | 400 | 400 |
| 24 | 600 | 400 | 600 |
| 25 | 600 | 600 | 200 |
| 26 | 600 | 600 | 400 |
| 27 | 600 | 600 | 600 |

A mock transduction (i.e., GSM transduction mixture only, 27 variants according to Table 7) and non-treated control will be included to understand CRISPR/Cas RNP toxicity when compared to the non-treated sample.

Also, we will try to substitute ethanol (permeabilizing agent of the DS) with isopropanol and DMSO. $0.8–2×10^5$ cells/well will be plated in 12-well culture plate the day before transduction. Briefly, 40 µL of modified DS with CRISPR/Cas RNP will be applied directly to the cells using an atomizer QA40 Atomizer equipped with P-80 and P-60 probes (Qsonica), incubated for 2 min at room temperature and washed for 30 sec with 200 µl of 0.5×PBS. Afterwards fresh culture medium will be added and cells will be incubated for further analysis.

A mock transduction control will be included to estimate CRISPR/Cas RNP toxicity when compared to the non-treated sample. All experiments will be performed in triplicate, 4 gRNAs will be used for RNP formation (AAVS1, CDK4 and HPRT1 and non-targeting control gRNA). Data will be presented as mean ± SD (n = 3), statistical analysis will be performed by one-way ANOVA.

**2.2.6. CRISPR/Cas RNP delivery to target cells using erythrocyte-based methods.** Delivery approaches that involve red blood cells can be used to efficiently deliver CRISPR/Cas RNP to target cells [68]. We will try to optimize both red blood cells (RBCs) loading with CRISPR/Cas RNP and red blood cells fusion with target cells. For RBCs loading the amount of CRISPR/Cas RNP will be optimized. To do this 0.5 ml packed RBCs (approximately $5×10^9$

**Table 8. Sucrose and PEG1000 concentrations optimization during standard osmotic lysis of pinocytic vesicles technique (48-well transduction plate layout).**

| | | | | Sucrose | | | | | | | |
|---|---|---|---|---|---|---|---|---|---|---|---|
| | | | | Concentration in Complete transduction mixture, mM | | | | | | | |
| | | | | 0 | 200 | 300 | 400 | 500 | 600 | | |
| | | | | 1 | 2 | 3 | 4 | 5 | 6 | 7 | 8 |
| **PEG1000** | Concentration in **Complete transduction mixture, %** | | A | - | - | - | - | - | - | - | - |
| | | 0 | B | - | | | | | | | - |
| | | 5 | C | - | | | | | | | - |
| | | 10 | D | - | | | | | * | | - |
| | | 20 | E | - | | | | | | | - |
| | | | F | - | - | - | - | - | - | - | - |

* - optimal combination according to [65]. Rows "A", "F" and columns #1, #8 will be filled with PBS to avoid edge effect.

cells) will be incubated with CRISPR/Cas RNP containing 62.5, 125, 250 or 500 μg of CRISPR/Cas protein in the presence of 400 μg HSA. Subsequently 2 mL of mQ water will be added to cause cell lysis and suspension will be incubated 45 min at room temperature. To seal RBCs 250 μL of 10×PBS will be added and the resulting RBC "ghosts" will be sedimented by centrifugation and washed. Afterwards, CRISPR/Cas RNP loaded RBC "ghosts" will be fused with $2.5\times10^7$ target cells in culture media containing of 12.5%, 25% or 50% (1–5 min at 25°C). Then this mixture will be slowly diluted with appropriate culture media and cells will be incubated for further analysis.

All experiments will be performed in triplicate, 4 gRNAs will be used for RNP formation (AAVS1, CDK4 and HPRT1 and non-targeting control gRNA). Data will be presented as mean ± SD (n = 3), statistical analysis will be performed by one-way ANOVA.

## 2.3. Viability assessment (post-transduction recovery)

Viability of transduced cells will be assessed using PrestoBlue™ HS Cell Viability Reagent (Thermo Fisher Scientific) or PrestoBlue™ Cell Viability Reagent (Thermo Fisher Scientific) according to manufacturer's recommendations. At each time-point (4-, 8-, 24-, 48-, 72- and 96-hours post-transduction) 1/10th volume of PrestoBlue™ Reagent will be added directly to cells in culture plates. Plates will be incubated for 30 minutes at 37°C in a cell culture incubator and fluorescence (excitation wavelength of 560 nm, range is 540–570 nm; emission of 590 nm, emission range is 580–610 nm) will be read on Varioskan™ LUX multimode microplate reader (Thermo Fisher Scientific). The percentages of cell viability in CRISPR/Cas RNP transduced cells, a "% Normalization" step will be carried out (the data will be normalized to 100% viable cells assuming that the signal obtained from the non-treated control wells corresponded to 100% viable cells). Also, the data from each experimental point will be analyzed using GraphPad Prism™ 9 Software (GraphPad Software).

After viability assessment culture media with PrestoBlue® Reagent will be removed and replaced with growth medium for further proliferation.

## 2.4. Protein delivery and genome editing efficacy assessment

**2.4.1. Delivery efficacy assessment using CellInsight™ CX5 High Content Screening (HCS) Platform.** Post-transfection efficiency of eGFP-tagged CRISPR/Cas RNP delivery will be assessed using CellInsight™ CX5 High Content Screening (HCS) Platform (Thermo Fisher Scientific) where NucBlue™ Live ReadyProbes™ Reagent (Thermo Fisher Scientific) will act as nuclear counterstain and Wheat Germ Agglutinin, Alexa Fluor™ 647 Conjugate (Thermo

Fisher Scientific)–as plasma membrane counterstain. At each time-point (4-, 8-, 24-, 48-, 72- and 96-hours post-transduction) NucBlue® Live reagent (2 drops/mL) and wheat germ agglutinin conjugate (1–10 μg/mL) will be added to the wells of culture plates and incubated for 10–30 minutes. Cells will be then imaged on CellInsight™ CX5 with the following protocol settings:

- Objective 10×;

- 2×2 binning;

- 4 channels for image acquisition (brightfield; blue; green; far-red).

Nine fields will be analyzed for each well. After CellInsight™ CX5 run will be complete obtained data will be exported to "CSV" file. Total intensity (TotalIntenCh3) gained from channel 3 (green) will reflect CRISPR/Cas RNP delivery efficacy. Obtained data will be analyzed using GraphPad Prism™ 9 Software.

**2.4.2. Delivery efficacy assessed via intracellular staining.** Moreover, non-eGFP-tagged CRISPR/Cas RNP delivered to adherent cell lines using the abovementioned delivery methods will be detected at the day of transduction via intracellular staining. Intracellular staining will be performed using Recombinant Alexa Fluor® 488 Anti-CRISPR-Cas9 antibody [EPR18991] (ab215239, Abcam) or Recombinant Anti-CRISPR-Cas9 antibody [EPR19633] (ab202657, Abcam) with Goat anti-Rabbit IgG (H+L) Cross-Adsorbed Secondary Antibody, PE (P2771MP, Thermo Fisher Scientific) secondary antibody staining. NucBlue™ Live ReadyProbes™ Reagent (Thermo Fisher Scientific) will act as nuclear counterstain and Wheat Germ Agglutinin, Alexa Fluor™ 647 Conjugate (Thermo Fisher Scientific)–as plasma membrane counterstain.

Briefly, the day before experiment 0.5–1×10⁵ cells will be seeded on Cell Culture Slide (30114, SPL). On the day of experiment cells will be transduced with CRISPR/Cas RNP. Intracellular staining will be performed 1-, 2- and 4-hours post-transduction. Cells will be fixed with 100% methanol for 5 minutes and subsequently permeabilized with 0.1% Triton X-100 for 5 minutes. Cells will be then blocked with 1% BSA in 0.1% PBS-Tween for 1 hour and incubated overnight at +4˚C with ab215239 at 1/1000 dilution (or incubated overnight at +4˚C with ab202657 at 1/100 dilution, washed trice in 0.1% PBS-Tween for 5 minutes and incubated with P2771MP). Then cells will be washed trice with 0.1% PBS-Tween for 5 minutes. NucBlue® Live reagent (2 drops/mL) and wheat germ agglutinin conjugate (1–10 μg/mL) will be added to the wells of culture plates and incubated for 10–30 minutes. Finally, cells will be mounted with SlowFade Diamond Antifade Mountant (S36967, Thermo Fisher Scientific). Visualization will be performed on the EVOS® FL Auto Imaging System. Produced images will be used to analyze intracellular localization of CRISPR/Cas RNP.

**2.4.3. Cytotoxicity assay.** Cytotoxicity will be assessed using the pre-installed Multiparameter Cytotoxicity BioApplication (CellInsight™ CX5), where changes in nuclear size/morphology, membrane permeability and lysosomal mass/pH will be investigated.

Non-eGFP-tagged CRISPR/Cas RNP will be delivered to target cells using the abovementioned delivery methods. 4-, 8-, 24-, 48-, 72- and 96-hours post-transduction 0.2 to 5 μg/mL Hoechst 33342 (Thermo Fisher Scientific), 100 nM to 5 μM TO-PRO™-3 Iodide (642/661) (Thermo Fisher Scientific) and 1 μM LysoSensor™ Green DND-189 (Thermo Fisher Scientific) will be added to the wells of culture plates and incubated for 30–60 minutes. The cells will then be fixed using formaldehyde, washed with 0.1% PBS-Tween and images will be collected using Multiparameter Cytotoxicity BioApplication on CellInsight™ CX5. Percent of cells with nuclear size/morphology changes will be counted from channel 2 (blue); cell membrane permeability will be estimated from channel 3 (far-red) and lysosomal mass–from channel 4

(green). Obtained data will reflect presence or absence of CRISPR/Cas RNP cytotoxicity. Statistical analysis will be performed using GraphPad Prism™ 9 Software.

**2.4.4. CRISPR/Cas RNP intracellular half-life.** eGFP-tagged CRISPR/Cas RNP intracellular half-life will be assessed using xCELLigence RTCA eSight (Agilent). Target cells will be seeded at the density of $6\times10^3$/well in a company-provided electronic 96-well microplate and cultured overnight. The next day, cells will be transduced eGFP-tagged CRISPR/Cas RNP and stained with 2 mM of the fluorescently labeled Caspase-3 dye (ACEA Biosciences) which is a marker for early apoptotic events. The cell index based on impedance values will be measured for up to 96 h. Also, four images per well will be taken by the eSight machine at a 10× magnification every hour using the GFP and DAPI channels. All signal-positive cells will be counted and plotted by the RTCA Software as object counts per well over time.

**2.4.5. Editing efficiency assessment using Digital PCR technique.** What is more, editing efficiency will be assessed using Digital PCR technique (QuantStudio™ 3D Digital PCR Instrument, Thermo Fisher Scientific). Post-transduction (4-, 8-, 24-, 48-, 72- and 96-hours) cells will be harvested and genomic DNA will be extracted using Blood and Tissue kit (Qiagen). The concentration of the obtained DNA samples will be measured on a Qubit 2.0 instrument using a Qubit dsDNA HS Assay Kit (Thermo Fisher Scientific).

Amplification of target regions that will undergo editing (AAVS1, CDK4 and HPRT1) will be performed and the resulting PCR products (300–500 bp) will be purified using SpeedBead Magnetic Carboxylate Modified Particles (GE Healthcare) and barcoded. Final libraries will be pooled in an equimolar ratio and the resulting pools will be purified using SpeedBead Magnetic Carboxylate Modified Particles (GE Healthcare). NGS (amplicon sequencing) will be performed on Illumina MiSeq (Illumina). The InDels will be identified by bioinformatics analysis. Oligonucleotides and probes (sensitive and insensitive to InDels) required for Digital PCR will be designed according to locations of the InDels found [69].

dPCR reactions will be prepared using (i) QuantStudio™ 3D Digital PCR Master Mix v2 (Thermo Fisher Scientific), (ii) oligonucleotides and probes (sensitive and insensitive to InDel) Assay(s) and (iii) DNA sample (200–2,000 copies/μL) and loaded onto a QuantStudio™ 3D Digital PCR 20K Chip v2. PCR will be performed using the ProFlex™ 2×Flat PCR System and chip will be read on the QuantStudio™ 3D Digital PCR Instrument. The results will be analyzed using the QuantStudio™ 3D Analysis Suite™ Software.

**2.4.6. Scalability assessment.** Assessing scalability has been identified as a fundamental step in any scaling up process. After optimal CRISPR/Cas RNP transduction conditions for each method will be defined experiments on scaling up will be performed. Usually, the recommended dosage of T-cell therapy for adult patients with body weight above 50 kg is 0.1–$2.5\times10^8$ cells. Prior to transduction cells will be grown in Corning® HYPERFlask® (adherent cells) or in 1000 mL Erlenmeyer Flask (suspension cells). Cells will be transduced with appropriate amount of CRISPR/Cas RNP pre-mixed with corresponding reagent (Lipofectamine™ CRISPRMAX™, TransIT-X2®, branched PEI, iTOP, GSM and other transduction mixtures).

For scaling-up CRISPR/Cas RNP electroporation 4D-Nucleofector™ System will be equipped with 4D-NucleofectorTM LV Unit. This unit allows closed, large-scale transfection of up to $1\times10^9$ cells. While, Neon™ electroporation will be scaled-up on CliniMACS Electroporator module. Cells will be electroporated under previously selected conditions (solution and program).

CRISPR/Cas RNP delivery will be analyzed in aliquots of transduced cells using CellInsight™ CX5 (refer to par. 4.1). Viability of cells and editing efficacy will be assessed as described previously (refer to par. 3 and 4.5 respectively). Parameters of large-scale CRISPR/Cas RNP delivery will be compared with those obtained earlier for small-scale transductions.

**2.4.7. Cost-effectiveness assessment.** Cost-effectiveness of each CRISPR/Cas RNP delivery method will be evaluated according to (i) amount of CRISPR/Cas RNP used for delivery, (ii) the price of all components used for delivery (solutions, cartridges, reagents and so on), (iii) the price of each run (launch) of equipment used for transduction, and (iv) operator hands-on time.

**2.4.8. IP rights analysis.** IP status of used transduction reagents will be clarified on https://www.thermofisher.com/ (Lipofectamine™ CRISPRMAX™), https://www.mirusbio.com/ (TransIT-X2®) and on https://www.lens.org/ (patents on iTOP, GSM and other protein delivery techniques).

## 2.5. Data analysis

Data analysis and graphing will be performed using GraphPad Prism™ 9 Software (GraphPad Software).

## 3. Discussion

Delivery of CRISPR/Cas RNPs still remains the biggest bottleneck to somatic-cell genome editing. Many efforts are made to develop efficient CRISPR/Cas RNP delivery methods that will not affect viability of target cell dramatically. Emerging strategies include advances in nanoparticle- and cell-based delivery methods [70] as well as approaches that involve red blood cells [71] and nanowires [72, 73]. But still, we need to find the most efficient, scalable and cost-effective method with potential for clinical use. From the first sight, some methods selected for this study are research-use only and may be useful in the field of functional genomics, cell screening procedures, cell line optimization processes and so on. Others–have potential in scaling-up and therefore can be used in therapy. The main goal of this future study is to assess whether some of these methods can be scaled-up for development of genome editing-based cell therapy products.

The main criteria to be assessed will be:

- protein delivery and genome editing efficacy;

- viability of target cells after delivery (post-transduction recovery);

- scalability of delivery process;

- cost-effectiveness of the delivery process;

- intellectual property rights.

The efficacy of the delivery method will be considered acceptable if the number of successfully edited cells will rich 60–80% for adherent cell lines, 40–60% for suspension cells and 40–80% for primary human cells. The viability of target cells after delivery should be at least 70–80%.

The scalability of the delivery process is very important when we are talking about primary human cells as they are target cells for development of genome editing-based cell therapy products while adherent and suspension cell lines are of potential interest only for functional genomics, cell screening procedures, cell line optimization processes and so on.

If the final selected protocol could not meet all the criteria mentioned above, the scalable one with the best indexes in protein delivery and genome editing efficacy, post-transduction recovery should be preferably considered for further usage.

Commercially available transfection reagents offer different options for CRISPR/Cas RNP delivery. At the moment, non-liposomal polymeric system, such as TransIT-X2®, and lipid

nanoparticle transfection system (Lipofectamine™ CRISPRMAX™) are the most popular for CRISPR/Cas RNP delivery. Both transfection systems promise superior transfection in various types of target cells. These transfection reagents have already demonstrated the ability to efficiently deliver CRISPR/Cas RNPs in various cell types including iPSC, mESC, N2A, CHO, A549, HCT116, HeLa, HEK293, etc. Moreover, both reagents promise low cell toxicity resulting in excellent viability post-transfection rates and cost savings. TransIT-X2® and Lipofectamine™ CRISPRMAX™ can be easily scaled up and are high throughput friendly according to the manufacturer (e.g., 115.1 µL of Cas9 Plus™ Reagent and 138.2 µL CRISPRMAX™ reagent will be sufficient enough to transfect up to $1.1 \times 10^7$ cells). Lipofectamine™ CRISPRMAX™ and TransIT-X2® are products of Thermo Fisher Scientific and Mirus Bio LLC respectively and may be covered by one or more Limited Use Label Licenses.

Branched polyethyleneimine (BPEI) is a polymer with repeating units composed of ethylene diamine groups used for transfection of DNA, RNA and proteins. BPEIs contain primary, secondary and tertiary amino groups. Complexes formed with BPEI have a robust surface that is highly positively charged. This transfection reagent is not yet well characterized in terms of CRISPR/Cas RNP delivery, there is no data on its toxicity. Presumably, BPEI transfections can be easily scaled up and there would be no need for licensing. BPEI has a large potential in CRISPR/Cas RNP delivery and needs further investigations.

There is a little data on CRISPR/Cas RNP delivery to different cell types via chemical membrane permeabilization (for example, iTOP, GSM, reversible permeabilization, etc.). It seems to be scalable, but cell toxicity needs further studies. All reagents used in chemical membrane permeabilization need to be GMP approved to be used in clinical practice and drug development. These methods are very promising as they are relatively cheap and do not require special equipment.

At this moment electroporation is considered the gold standard for CRISPR/Cas RNP delivery to target cells. For example, clinically approved CliniMACS Prodigy® Instrument (Miltenyi Biotec) is equipped with CliniMACS Electroporator module. This module expands the possibilities of automated cell processing and enables flexibility in large-scale transfection of various cell types, e.g., primary human cells. The electroporation step is integrated into the cell manufacturing workflow and takes place after washing and rebuffering on the CliniMACS Prodigy®. During this process cells are transferred to the electroporation cuvette, which is a component of the single-use CliniMACS Electroporation Tubing Set. The cell suspension is divided into smaller samples and mixed with transfection material such as nucleic acids or CRISPR/Cas RNPs. After electroporation, cells are transferred back to the CliniMACS Prodigy for downstream cell processing.

Transduction with CRISPR/Cas RNPs also can be integrated into CliniMACS Prodigy® workflow. The only difference from the CliniMACS Prodigy TCT process is that instead of the preparation containing the viral particles, CRISPR/Cas RNPs in the transduction buffer will be added to the system.

High-throughput electroporation is a well-known technique, but CRISPR/Cas RNP delivery may be followed by cell loss (>50%) and changes in cell phenotype and function. Moreover, challenges to scale from benchtop to clinical scale may appear as special equipment is required.

As there is a little or no data about the optimal delivery method to introduce CRISPR/Cas RNP to clinically relevant target cells, our study will be of potential interest. This will be the first extensive comparative study of popular current methods and protocols of CRISPR/Cas RNP delivery to human cell lines and primary cells. All protocols will be optimized and characterized. Some methods will be considered 'research-use only', others - will be recommended for scaling and application in the development of cell-based therapies.

## Author Contributions

**Conceptualization:** Marina A. Tyumentseva, Aleksandr I. Tyumentsev.

**Formal analysis:** Marina A. Tyumentseva, Aleksandr I. Tyumentsev.

**Funding acquisition:** Vasiliy G. Akimkin.

**Investigation:** Marina A. Tyumentseva, Aleksandr I. Tyumentsev.

**Methodology:** Marina A. Tyumentseva, Aleksandr I. Tyumentsev.

**Project administration:** Marina A. Tyumentseva, Aleksandr I. Tyumentsev.

**Supervision:** Vasiliy G. Akimkin.

**Writing – original draft:** Marina A. Tyumentseva, Aleksandr I. Tyumentsev.

**Writing – review & editing:** Vasiliy G. Akimkin.

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
