## [Decision Letter · Decision Letter 0]

20 Aug 2021

PONE-D-21-23919

Protocol for assessment of the efficiency of CRISPR/Cas RNP delivery to different types of target cells

PLOS ONE

Dear Dr. Tyumentseva,

Thank you for submitting your manuscript to PLOS ONE. After careful consideration, we feel that it has merit but does not fully meet PLOS ONE’s publication criteria as it currently stands. Therefore, we invite you to submit a revised version of the manuscript that addresses the points raised during the review process.

We look forward to receiving your revised manuscript.

Kind regards,

Zhiqiang Wen, Ph.D.

Academic Editor

PLOS ONE

Journal Requirements:

Reviewers' comments:

Reviewer's Responses to Questions

**Comments to the Author**

1. Does the manuscript provide a valid rationale for the proposed study, with clearly identified and justified research questions?

Reviewer #1: Yes

Reviewer #2: Yes

2. Is the protocol technically sound and planned in a manner that will lead to a meaningful outcome and allow testing the stated hypotheses?

Reviewer #1: Partly

Reviewer #2: Partly

3. Is the methodology feasible and described in sufficient detail to allow the work to be replicable?

Reviewer #1: Yes

Reviewer #2: Yes

4. Have the authors described where all data underlying the findings will be made available when the study is complete?

Reviewer #1: Yes

Reviewer #2: Yes

5. Is the manuscript presented in an intelligible fashion and written in standard English?

Reviewer #1: Yes

Reviewer #2: Yes

6. Review Comments to the Author

You may also provide optional suggestions and comments to authors that they might find helpful in planning their study.

Reviewer #1: This work describes a detailed protocol for screening an optimal condition for delivery of RNP into different cells. But this manuscript is poorly written, including too many repeated sentences and improper vocabulary usages. Lack of novelty also extremely hampers the reader's enthusiasm. Therefore, I recommend accepting this manuscript for PLOS ONE publication after major revisions as below.

1.The repeated sections should be carefully revised, e.g. Line 227, Line 297, Line 314.

2.Authors should indicate or explain the meaning of the last column and row in Table 4-6. Is it possible to increase the concentration of the reagents to such a high level?

3.More discussion about the influence on the efficiency, viability, large-scale production under different conditions including transfection reagents, electroporation, and so on.

4.If the final selected protocol could not meet all the criteria, which ones should be preferably considered?

5.What’s the difference between adherent cells and suspension cells in the protocol design? Similar protocols are suggested to be put together.

6.The part in CD4/CD8 cells is also similar to that of CD34 cells. How to design a specific protocol based on the different types of cells?

Reviewer #2: In the study, the author developed several methods and protocols of CRISPR/Cas RNP delivery to target cells to provide reference for the genome editing in clinical practice and drug delvelopment. The authors still need to address the following issues.

1. In the introduction, please describe current methods used for CRISPR/Cas delivery by citing relevant literatures, and then explain your own innovation.

2. Some methods were not detailed enough, such as the electrical transfer conditions, cytotoxicity test procedures, and the amount of CRISPR/Cas RNP added...

3. Editing efficiency is also an important criteria to measure delivery efficiency except for the five items mentioned in the paper.

4. It is mentioned in the paper that scalability of delivery process, cost-effectiveness of the delivery process and intellectual property rights are also evaluation criteria, but they are not evaluated in the method. Please supply it.

7. PLOS authors have the option to publish the peer review history of their article (what does this mean?). If published, this will include your full peer review and any attached files.

Reviewer #1: No

Reviewer #2: No

---

## [Author Response · Author response to Decision Letter 0]

7 Oct 2021

Response to Reviewer #1 comments:

Dear Reviewer,

We want to thank you for the thoughtful feedback. We have incorporated all of your comments to our revision, tried to eliminate most of repeated sections and added some points to discussion. We also provide a point-by-point reply to all of your comments.

Comment 1:

We’ve carefully revised the repeated sections and put them together.

Comment 2:

We’ve explained the meaning of the last column and row in Table 4-6.

We suggest that there is no need to elevate concentration of glycerol above 10%, glycine – above 135 mM, betaine – above 270 mM and proline – above 180 mM, as these values exceed the optimal ones 1.8-45 times. Whether it is possible to increase the concentration of the reagents to such a high level or not we will see when we will be performing viability tests. The main aim is to optimize delivery conditions and we need to evaluate boundary (extreme) parameters.

Comment 3:

More discussion about potential efficiency, viability, large-scale production under different conditions including transfection reagents, electroporation, and so on was added.

Comment 4:

If the final selected protocol could not meet all the criteria mentioned above, the scalable one with the best indexes in protein delivery and genome editing efficacy, post-transduction recovery should be preferably considered for further usage. We’ve also added this phrase to discussion.

Comment 5:

All similar protocols were put together.

Comment 6:

Protocols for CD4/CD8 cells and CD34 cells also were put together. The main goal of our future study is optimization of CRISPR/Cas delivery, and we hope this will help us to design a specific protocols for CD4/CD8 cells and CD34.

On behalf of authors, I want to thank Reviewer #1. All comments were very helpful.

Best regards,

Marina Tyumentseva.

 

Response to Reviewer #2 comments:

Dear Reviewer,

On behalf of all co-authors, I thank you for your time and efforts on revising the manuscript. We now submit the revised text. We’ve carefully revised our manuscript according to your suggestions.

Comment 1:

In the introduction, we’ve described current methods used for CRISPR/Cas delivery by citing relevant literatures. Our innovation is that we will perform the first extensive comparative study of popular current methods and protocols of CRISPR/Cas RNP delivery to human cell lines and primary cells. Also, we will perform optimization of these methods and will try to offer new approaches in CRISPR/Cas RNP delivery (e.g., RBCs ‘ghosts’). Optimized and intermediate methods will be systematically characterized (delivery efficacy, editing efficacy, post-delivery viability, CRISPR/Cas RNP half-life and toxicity, etc.).

Comment 2:

We’ve tried to add more details to methods mentioned in our manuscript. Electrical transfer conditions were indicated for Neon Transfection System, while for 4D-Nucleofector system we’ve indicated only programs (as 4D-Nucleofector is a closed system). Also, we’ve added more description to cytotoxicity, viability, delivery and genome-editing efficacy tests. For each delivery procedure the amount of CRISPR/Cas RNP was added.

Comment 3:

We completely agree that editing efficiency is an important criterion. Editing efficiency was added to Fig 1 and text along side with delivery efficiency and other criteria.

Comment 4:

We’ve added chapters “4.6. Scalability assessment”, “4.7. Cost-effectiveness assessment” and “4.8. IP rights analysis” regarding evaluation of such criteria as scalability of delivery process, cost-effectiveness of the delivery process and intellectual property rights to our manuscript.

We want to thank Reviewer #2 as all suggestions helped us to revise our manuscript.

Best regards,

Marina Tyumentseva.

---

## [Decision Letter · Decision Letter 1]

27 Oct 2021

Protocol for assessment of the efficiency of CRISPR/Cas RNP delivery to different types of target cells

PONE-D-21-23919R1

Dear Dr. Tyumentseva,

We’re pleased to inform you that your manuscript has been judged scientifically suitable for publication and will be formally accepted for publication once it meets all outstanding technical requirements.

Kind regards,

Zhiqiang Wen, Ph.D.

Academic Editor

PLOS ONE

Additional Editor Comments (optional):

Reviewers' comments:

Reviewer's Responses to Questions

**Comments to the Author**

1. Does the manuscript provide a valid rationale for the proposed study, with clearly identified and justified research questions?

Reviewer #1: Yes

Reviewer #3: Yes

2. Is the protocol technically sound and planned in a manner that will lead to a meaningful outcome and allow testing the stated hypotheses?

Reviewer #1: Yes

Reviewer #3: Yes

3. Is the methodology feasible and described in sufficient detail to allow the work to be replicable?

Reviewer #1: Yes

Reviewer #3: No

4. Have the authors described where all data underlying the findings will be made available when the study is complete?

Reviewer #1: Yes

Reviewer #3: Yes

5. Is the manuscript presented in an intelligible fashion and written in standard English?

Reviewer #1: Yes

Reviewer #3: No

6. Review Comments to the Author

You may also provide optional suggestions and comments to authors that they might find helpful in planning their study.

Reviewer #1: Authors have carefully revised the manuscript and all the comments have been addressed. Therefore, I recommend accepting this manuscript for PLOS ONE publication.

Reviewer #3: The author has answered and addressed all my questions. Current manuscript could be considered to publish.

7. PLOS authors have the option to publish the peer review history of their article (what does this mean?). If published, this will include your full peer review and any attached files.

Reviewer #1: No

Reviewer #3: No

---

## [Editor Report · Acceptance letter]

29 Oct 2021

PONE-D-21-23919R1 

Protocol for assessment of the efficiency of CRISPR/Cas RNP delivery to different types of target cells 

Dear Dr. Tyumentseva:

I'm pleased to inform you that your manuscript has been deemed suitable for publication in PLOS ONE. Congratulations! Your manuscript is now with our production department. 

Kind regards, 

on behalf of

Dr. Zhiqiang Wen 

Academic Editor

PLOS ONE